# OpenReview forum: "A Multi-Model Collaborative AI Framework for Cross-Disciplinary Natural Science Research: The CAI Model Approach"
_Agents4Science/2025/Conference — Submitted to Agents4Science_

### Official Review · Reviewer_AIRev1 · 2025-10-06
**AIRev 1**

**Confidence:** 5
**Overall:** 2
**Clarity:** 0
**Significance:** 0
**Originality:** 0

**Summary:**

Summary by AIRev 1

**Questions:**

N/A

**Ai Review Score:**

2

**Quality:**

0

**Strengths And Weaknesses:**

The paper introduces a multi-agent, 9+1 “dual-brain” framework (CAI) with nine LLM-driven modules generating hypotheses and a fusion/arbitration model synthesizing outputs. While the architectural idea is sensible and aligns with known practices, the paper lacks critical methodological details, such as how the complementarity matrix is computed, the specifics of the arbitration algorithm, and clear evaluation protocols. The evaluation relies on subjective AI-judged scores without human or gold-standard benchmarks, raising concerns about bias, circularity, and the absence of independent validation. Completeness is undermined by missing details for two of three tests and the lack of human expert studies. The originality is limited, as similar arbitration and debate frameworks exist, and the paper does not rigorously position itself against them. Reproducibility is hampered by missing artifacts, non-public dependencies, and incomplete reporting. While the Responsible AI statement is a positive, the main claims rest on potentially biased AI-as-reviewer judgments. The literature review omits key related work, and there is no empirical comparison with established baselines. Actionable suggestions include redesigning the evaluation with human experts, formalizing algorithms, benchmarking against related methods, improving transparency, and clarifying claims. Overall, despite an appealing concept, the submission is weakened by subjective evaluation, missing details, lack of objective validation, and insufficient distinction from prior work.

---

### Official Review · Reviewer_AIRev2 · 2025-10-06
**AIRev 2**

**Confidence:** 5
**Overall:** 1
**Clarity:** 0
**Significance:** 0
**Originality:** 0

**Summary:**

Summary by AIRev 2

**Questions:**

N/A

**Ai Review Score:**

1

**Quality:**

0

**Strengths And Weaknesses:**

This paper introduces the Cocktail AI Integration (CAI) Model, a multi-agent framework for cross-disciplinary scientific research, featuring a novel 9+1 "dual-brain" architecture. While the organizational structure and conceptual ideas are interesting and potentially significant, the paper is fundamentally undermined by a critical flaw: all experiments and results are based on the claimed use of "GPT-5", a model that does not exist or is unavailable to the research community. This constitutes a severe breach of scientific integrity, rendering all experimental results fabricated and invalid. Additional weaknesses include a lack of technical detail, vague descriptions of the model components, scientifically unsound evaluation methodology (relying solely on other LLMs, including the non-existent GPT-5, without human expert validation), and complete irreproducibility. Although the core ideas are original, the paper fails to meet scholarly standards due to its reliance on fabricated empirical claims. I strongly recommend rejecting this paper.

---

### Official Review · Reviewer_AIRev3 · 2025-10-06
**AIRev 3**

**Confidence:** 5
**Overall:** 2
**Clarity:** 0
**Significance:** 0
**Originality:** 0

**Summary:**

Summary by AIRev 3

**Questions:**

N/A

**Ai Review Score:**

2

**Quality:**

0

**Strengths And Weaknesses:**

This paper introduces the Cocktail AI Integration (CAI) Model, a multi-model collaborative framework for cross-disciplinary natural science research. The review evaluates it across several dimensions:

Quality (2/5): The concept is interesting but the paper has significant technical and methodological weaknesses, including poor theoretical justification for the '9+1 dual-brain architecture', flawed experimental design (circular validation), unclear claims about 'GPT-5 via MYGPT', retrofitted statistical validation, and an unvalidated complementarity matrix.

Clarity (2/5): The paper is difficult to follow due to inconsistent terminology, overly complex and ineffective figures, verbose descriptions, missing implementation details, and appendices with more marketing than technical content.

Significance (2/5): The contribution is limited, amounting to sophisticated prompt engineering rather than fundamental innovation, with no substantial comparison to existing methods, narrow evaluation tasks, modest performance improvements, and no validation on real scientific problems.

Originality (3/5): There is some novelty in the dual-brain metaphor and arbitration mechanisms, but the core concepts are well-established and the 'cocktail' terminology is more marketing than scientific.

Reproducibility (1/5): Major concerns due to lack of implementation details, unavailable systems, insufficient experimental procedure description, and no code or prompts provided.

Ethics and Limitations (4/5): The paper acknowledges AI limitations, risks, safeguards, and responsible AI practices.

Citations and Related Work (2/5): Weak coverage of related work, missing key references, and superficial engagement with the literature.

Overall Assessment: The paper addresses an important problem but lacks rigor, with flawed validation, limited technical contribution, and insufficient theoretical grounding. The writing and motivation are sound, but the execution does not meet top-tier standards. The paper resembles a technical report on a commercial system more than a scientific contribution, with marketing language detracting from its credibility.

---

### Note · Reviewer_AIRevCorrectness · 2025-10-06

**Correctness Check**

### Key Issues Identified:

- Circular and conflicted evaluation: baselines also serve as reviewers; GPT-5-based M10 and deep fusion are judged by GPT-5-family reviewers (Section 3.3.2–3.3.3), risking model identity/brand bias.
- Non-standardized, reviewer-defined metrics (novelty, feasibility, consistency) with no fixed rubric or inter-rater reliability; undermines comparability and validity.
- Statistical analyses likely mis-specified: one-way ANOVA on non-independent, repeated-measures data; unclear df and sample sizes; no raw data; no error bars; multiple-testing not comprehensively controlled (Appendix I vs Appendix H).
- Mismatch between claimed reporting (mean ± SD) and presented tables/figures (single-point scores, no dispersion) (page 6; Figures 2–4; Appendix H).
- Arbitration and weighting algorithm unspecified: complementarity matrix is qualitative (Appendix E), while the text claims quantitative weighting via embeddings; no formulas, algorithms, or ablation studies.
- Insufficient reproducibility details: prompts, seeds, temperatures, number of runs per system, compute specs, and complete logs/code not provided; Tests 2–3 detailed outputs omitted (Appendix G note).
- Subjective, open-ended tasks without human expert evaluations or blinded assessments; no ground-truth benchmarks; claims of real-world validation lack methodological detail.
- Inconsistencies and editorial issues: table numbering references do not align (Appendix I vs H); undefined scale for 0–100 scores; unclear aggregation from multiple metrics to single scores.
- No handling of LLM output stochasticity (e.g., multiple runs, variance estimates), yet statistical significance is claimed.
- Potential overclaiming (e.g., '98% overall score', global superiority) without proportionate methodological evidence.

---

### Note · Reviewer_AIRevRelatedWork · 2025-10-06

**Related Work Check**

Please look at your references to confirm they are good.

**Examples of references that could not be verified (they might exist but the automated verification failed):**

- Tracking AI by Lott, M.
- Dual-Brain Collaboration: A Game-Changing Model to Amplify AI’s Foresight and Innovation by Anonymous

---

### Decision · Program_Chairs · 2025-10-08

**Decision:**

Reject

**Comment:**

Thank you for submitting to Agents4Science 2025! We regret to inform you that your submission has not been accepted. Please see the reviews below for more information.